# LEARNING SURROGATE LOSSES

## ABSTRACT

The minimization of loss functions is the heart and soul of Machine Learning. In this paper, we propose an off-the-shelf optimization approach that can seamlessly minimize virtually any non-differentiable and non-decomposable loss function (e.g. Miss-classification Rate, AUC, F1, Jaccard Index, Mathew Correlation Coefficient, etc.). Our strategy learns smooth relaxation versions of the true losses by approximating them through a surrogate neural network. The proposed loss networks are set-wise models which are invariant to the order of mini-batch instances. Ultimately, the surrogate losses are learned jointly with the prediction model via bilevel optimization. Empirical results on multiple datasets with diverse real-life loss functions compared with state-of-the-art baselines demonstrate the efficiency of learning surrogate losses.

## 1 INTRODUCTION

In reality, a large set of loss functions cannot be directly minimized by gradient-based methods because they are either piece-wise continuous, non-differentiable, or non-decomposable (Zhang et al., 2018). For example, binary classification models are often evaluated using the miss-classification rate (MCR), Area under the ROC curve (AUC), F1 measure (F1), Jaccard Similarity Index (JAC), Average Precision (AP), Equal Error Rate (EER), or the Mathew Correlation Coefficient (MCC). On the other hand, learning-to-rank models are measured through the Normalized Discounted Cumulative Gain (NDCG), or the Mean Average Precision (MAP). To illustrate the point, Figure 1 shows the challenging surfaces of common binary classification losses. Unfortunately, there exists no tractable omni-solver so far, i.e. no off-the-shelf optimization strategy to train prediction models for the aforementioned category of loss functions. It is worth pointing out that non-gradient-based approaches, such as evolutionary computing, are intractable from a runtime perspective.

While there exists a plethora of methods that tackle various aspects of particular losses, there is no single gradient-based method that can optimize any loss in a seamless manner. Researchers have so far focused on deriving smooth surrogate relaxations (approximations) to the true loss functions (Berman et al., 2018; Eban et al., 2017), in a way that first- and second-order optimization techniques can benefit from the derivative of the surrogates with respect to the parameters of prediction models. For instance, the widely-applied cross-entropy loss is a surrogate relaxation of the miss-classification rate. However, these explicit relaxations are hand-crafted individually for each loss and do not generalize to other losses.

In this paper, we present the first off-the-shelf optimizer for arbitrary loss functions. In contrast to the related work, this work proposes a new perspective on minimizing loss functions, by defining surrogate losses as meta-level neural networks that approximate the desired true non-differentiable losses. Our method does not need the gradient information of the true loss with respect to the parameters of the prediction model and treats the loss as a black-box function. In addition, we introduce a set-based surrogate network that computes the loss over the training set, being invariant to the order of instances, in order to accurately handle non-decomposable losses.

The surrogate learning problem is formalized as a bilevel programming task (Colson et al., 2007; Franceschi et al., 2018) that is trained through a concurrent optimization algorithm. This paper shows that universal surrogates, which are trained without paying attention to a specific dataset, are sub-optimal compared to surrogates learned in a per-dataset manner. Results on nine datasets demonstrate that learning surrogates produces more accurate prediction models than state-of-the-art baselines with regard to diverse loss functions.

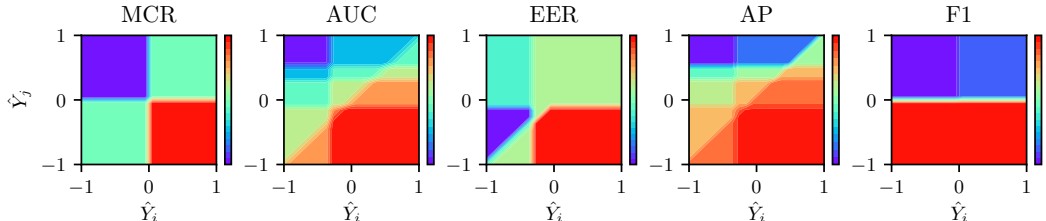

Figure 1: The surfaces of five binary classification losses derived by perturbing the predictions $\hat{Y}_i, \hat{Y}_j$ of two instances inside a random mini-batch $\hat{Y} \in \mathbb{R}^{10}, Y \in \{0, 1\}^{10}$, with true targets $Y_i = 0, Y_j = 1$. AUC, AP and F1 are converted to losses via $1 - x$, while the positive class for MCR, EER and F1 is estimated as $\hat{Y} \geq 0$.

## 2 RELATED WORK

Due to the non-differential and non-continuous nature of most real-life losses, early works deployed the proxies of the miss-classification rate (e.g. cross-entropy) as universal proxy losses, despite their sub-optimal performance (Cortes & Mohri, 2004). Subsequent approaches relied on designing smooth relaxations of the losses. As an example, the pairwise ranking loss is a common surrogate of the AUC measure (Gao & Zhou, 2015; Chen et al., 2009). On the other, hand, the F-measure is another typical loss that cannot be optimized directly due to its non-decomposable nature over instances. The initial papers tackling F1 focused instead on the empirical utility maximization paradigm (Ye et al., 2012). Later on, researchers addressed F1 by optimizing the hyper-parameters of standard binary classifiers, either the cost-sensitive weights of the classification loss (Puthiya Parambath et al., 2014), or the threshold of the estimated target values (Lipton et al., 2014; Koyejo et al., 2014). Nevertheless, the non-decomposable trait of F1 remains unresolved (Zhang et al., 2018) and recent works have explored directions to improve hyper-parameter tuning with tighter bounds (Bascol et al., 2019).

Instead of relying on explicit surrogates, another research direction handles non-convex losses by means of the *direct loss* method (Hazan et al., 2010), which minimizes a surrogate loss by embedding the true loss as a correction term. This method was recently extended to optimize neural networks (Song et al., 2016). It assumes the loss can be decomposed into per-instance sub-losses and the authors derived an explicit decomposition of the average precision (Song et al., 2016). Unfortunately, per-instance dis-aggregations are not trivially feasible in other cases (e.g. F1), making the direct loss optimization technique an impractical off-the-shelf option.

Two more recent papers have offered relaxation surrogates for non-decomposable losses, concretely the AUC and the Jaccard Index (known as Intersection over Union in the computer vision community). The first method defines relaxation forms for the building blocks of a confusion matrix (e.g. true positives, true negatives etc.) and combines the building block relaxations to create a final surrogate for losses like the AUC (Eban et al., 2017). However, this model does not handle cases where the loss is not expressible into confusion matrix blocks, for instance the Jaccard index. On the other hand, the second paper proposes the Lovasz soft-max as a smooth approximation to the Jaccard index (Berman et al., 2018), which is based on a generic decomposition of sub-modular (decomposable) losses for sets (Yu & Blaschko, 2015).

Besides that, a stream of recent papers has focused on meta-learning for loss functions. The "Learning to Teach" paradigm (Fan et al., 2018) proposes a meta-level teacher/controller that continuously updates the loss function for a prediction model based on its progress. The work has been recently extended to enable a gradient-based learning of the teacher/controller (Wu et al., 2018). However, this approach does not extend to non-decomposable loss functions which are defined over the full set of instances. A parallel stream of research has elaborated the concept of "Learning to Learn" (Li & Malik, 2017), or "Learning to Optimize" (Chen et al., 2017), which proposes to directly learn the amount of update values that are applied to the parameters of the prediction model. In the proposed framework a controller uses per-parameter learning curves comprised of the loss values and derivatives of the loss with respect to each parameters (Chen et al., 2017). This method suffers from two drawbacks that prohibit its direct applicability to arbitrary losses: i) for large prediction models it is computationally infeasible to store the learning curve of every parameter, and ii) there is no gradient information for non-differentiable losses.

An alternative approach towards learning loss functions analyzes the usage of discriminative adversarial networks (dos Santos et al., 2017). The idea focuses on discriminating the probability of a given target variable value into either being an estimated target or the true one, therefore acting as a form of surrogate loss (dos Santos et al., 2017). Last, but not least, our method shares similarities with the concept of error-critic model for function approximation (Pang & Werbos, 1998). In contrast to the prior work, we propose the first off-the-shelf optimization method that seamlessly minimizes any loss function. In Section 4.3 we empirically compare the proposed method against multiple state-of-the-art relaxations, with regards to minimizing popular binary classification losses, such as AUC, F1, Jaccard Index and the miss-classification rate.

## 3   SURROGATE LEARNING

Data mini-batches $(x, y) := \{(x_1, y_1), \ldots, (x_N, y_N)\}$ of $N$ instances each, with features $x \in \mathbb{R}^{N \times M}$ and ground-truth targets $y \in \mathcal{Y}^N$, are drawn from a dataset $\mathcal{D}$ with a sampling distribution $P_{\mathcal{D}}(x, y)$, where typically $P$ is the uniform distribution. The target domain can be binary $y \in \{0, 1\}^N$, or nominal among $C$ categories $y \in \{1, \ldots, C\}^N$. Ultimately, the purpose of a prediction model is to estimate a target variable $\hat{y}(x; \alpha) \colon \mathbb{R}^{N \times M} \to \mathbb{R}^N$, where the prediction model has parameters $\alpha$. The estimations $\hat{y} \in \mathbb{R}^N$ need to accurately match the given ground-truth target variable $y \in \mathcal{Y}^N$ with regards to a desired loss function $\ell(y, \hat{y}) \colon \mathbb{R}^N \times \mathcal{Y}^N \to \mathbb{R}$. Therefore, supervised learning focuses on computing the optimal parameters $\alpha^*$ that minimize the following objective.

$$\alpha^* = \arg\min_{\alpha} \quad \mathbb{E}_{(x,y) \sim P_{\mathcal{D}}(x,y)} \quad \ell(y, \hat{y}(x; \alpha)) \tag{1}$$

To minimize the aforementioned objective through first- or second-order optimization, it is necessary to define the gradients $\frac{\partial \ell}{\partial \alpha}$. Unfortunately, in most real-world cases the loss represents step functions that are only piece-wise continuous (MCR, F1, AUC, etc.). Therefore, the derivatives are either zero, or undefined at the function steps, which prohibits a direct optimization of these losses. For this reason, a smooth surrogate relaxations of the loss functions is used instead of the true loss. Arguably the most popular surrogate is cross-entropy, which is a relaxation of the miss-classification rate. Yet, such non-parametric relaxation functions are not trivially derivable when the loss is non-decomposable into per-instance components (e.g. F1, AUC), because such losses are defined as performance measures over an entire set of instances.

Instead of deriving one explicit hand-crafted function $\hat{\ell}$ for the surrogate of every loss function $\ell$, we propose a method that *parameterizes* and *learns* the surrogate for any demanded loss function through an off-the-shelve procedure. Neural networks are a good choice for parametrizing the surrogate loss $\hat{\ell}$ given their universal approximation capability (Hornik, 1991). However, the surrogate loss network must be a permutation-invariant set model, whose output must remain fixed given different orders of instances within a dataset mini-batch, i.e. $\hat{\ell}(y_1, \ldots, y_N, \hat{y}_1, \ldots, \hat{y}_N) = \hat{\ell}(y_{\pi(1)}, \ldots, y_{\pi(N)}, \hat{y}_{\pi(1)}, \ldots, \hat{y}_{\pi(N)})$ for any index permutation $\pi$. Our method uses the KolmogorovArnold representation theorem (Tikhomirov, 1991) and defines the surrogate loss of Equation 2 as a composition function $h$ of per-instance functions $g$. This type of aggregation was recently applied to learning neural networks over sets of instances (Zaheer et al., 2017).

$$\hat{\ell}(y, \hat{y}) = h\left(\frac{1}{N} \sum_{i=1}^{N} g(y_i, \hat{y}_i)\right) \tag{2}$$

where $g \colon \mathbb{R}^2 \to \mathbb{R}^Q$, and $h \colon \mathbb{R}^Q \to \mathbb{R}$ are deep forward neural networks. The first function $g$ extracts $Q$ latent error components for the predictions on each instance (e.g. latent representations of false positive, false negative, etc.), while the aggregation produces set-wise performance indicators (e.g. potentially latent representation of the count of true positives, false positives, error rate, etc.). The function $h$ creates nonlinear combinations of set-wise performance indicators to produce complex latent error metrics such as precision, recall, F1, etc.

It has been proven that with sufficient capacities for $g$ and $h$, any permutation-invariant set function (hence any loss) can be approximated via the Kolmogorov-Arnold (KA) representation theorem (Za-

heer et al., 2017). In particular, it is worth mentioning that due to being an instance of the KA superposition, the proposed surrogate model of Equation 2 benefits from existing error bound guarantees of such a representation when the functions $g$ and $h$ are deep neural networks (Montanelli & Yang, 2019). The remaining sections of this paper detail a novel method for optimizing surrogate losses.

### 3.1 UNIVERSAL SURROGATES

The first intuition is to learn the weights $\beta$ of surrogate network (a.k.a. the functions $g$ and $h$) in a universal manner by solving the objective of Equation 3. In other words, we can attempt to make any surrogate loss $\hat{\ell}$ behave as any true loss $\ell$ over the space of all possible batches of randomly-drawn true $y$ and estimated $\hat{y}$ targets. Focusing on binary classification losses, we sample the ground truth from a Bernoulli distribution, and respective estimated targets from a Gaussian distribution. Furthermore, the meta-loss $\mathcal{L} : \mathbb{R} \times \mathbb{R} \to \mathbb{R}$ measures the distance between the true and surrogate losses and is practically implemented as an $L_1$ norm.

$$\beta^{\text{Universal}} = \arg\min_{\beta} \quad \mathbb{E}_{y \sim \mathcal{B}(p)_{N \times 1}, \, \hat{y} \sim \mathcal{N}(\mu, \sigma^2)_{N \times 1}} \quad \mathcal{L}\left(\ell\left(y, \hat{y}\right), \hat{\ell}\left(y, \hat{y}\,;\,\beta\right)\right) \tag{3}$$

The challenge of learning a universal surrogate is in ensuring that the sampling hyper-parameters $(p, \mu, \sigma)$ would lead to drawing $(y, \hat{y})$ that match the specific target distribution of a concrete dataset. Therefore, we empirically found out that it is sub-optimal to universally relax the whole space of the true loss. Instead, as future sections will detail, it is more efficient to design a relaxation in a dataset-specific manner, which smoothens only the specific regions of the true loss which belong to the respective dataset-specific distributions of true and estimated target batches.

### 3.2 SURROGATE LEARNING AS BILEVEL PROGRAMMING

A different approach from the universal surrogate is to associate a surrogate loss to each dataset. In that manner, the surrogate creates a smooth relaxation only around the dataset-relevant regions of the target $(y, \hat{y}(x; \alpha))$ space. We propose to jointly optimize the prediction model parameters $(\alpha)$ and the surrogate loss parameters $(\beta)$ through the following optimization:

$$\alpha^* = \arg\min_{\alpha} \quad \mathbb{E}_{(x,y) \sim P_{\mathcal{D}}(x,y)} \quad \hat{\ell}\left(y, \hat{y}\left(x\,;\,\alpha\right)\,;\,\beta\right) \tag{4}$$

$$\beta^* = \arg\min_{\beta} \quad \mathbb{E}_{(x,y) \sim P_{\mathcal{D}}(x,y)} \quad \mathcal{L}\left(\ell\left(y, \hat{y}\left(x;\alpha\right)\right), \hat{\ell}\left(y, \hat{y}\left(x;\alpha\right)\,;\,\beta\right)\right) \tag{5}$$

The rationale is that Equation 4 optimizes the prediction model in order to minimize the surrogate loss. However, the surrogate loss should approximate the true loss, which is ensured by Equation 5.

Such interdependence can be addressed by treating both objectives of Equations 4 and 5 as a concurrent relationship, in a way that we can optimize them jointly and simultaneously. This dual objective is an instance of a bilevel programming problem (Colson et al., 2007; Franceschi et al., 2018). Algorithm 1 sketches the minimization procedure of the proposed surrogate learning. In an alternating fashion, the surrogate minimization is carried out for $K_\alpha$ steps (lines $4 - 7$), while the surrogate loss is updated for $K_\beta$ steps (lines $9 - 12$). To illustrate the mechanism, Figure 2 provides a minimalistic example of optimizing jointly the bilevel programming objectives for training a single-parameter model on a single-feature binary dataset. The plots show the effect of minimizing the surrogate (magenta), as the parameter $\alpha$ is updated towards the minimum of the surrogate. At the same time, we observe that the surrogate is updated to match the true miss-classification-rate loss (cyan) at the current parameter value. The model converges close to the optimal true loss after applying the steps of Algorithm 1.

### 3.3 NOTE ON CONVERGENCE

It is possible to analyze the convergence of Algorithm 1 following existing proofs for bilevel programming (Revalski & Zhivkov, 1993; Franceschi et al., 2018), since Equations 4-5 can be rewritten

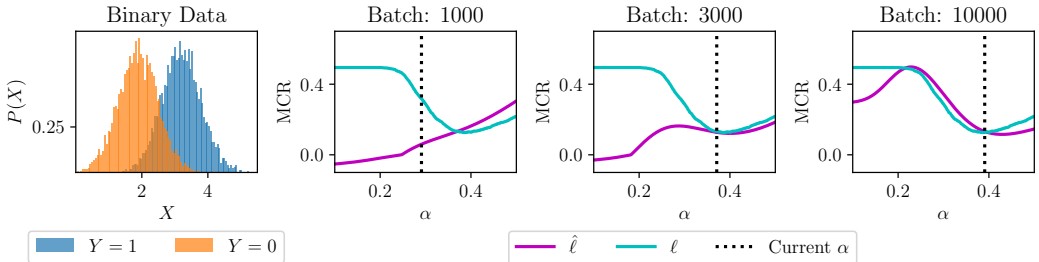

Figure 2: A single-parameter prediction model $\hat{y}(x, \alpha) = \alpha x - 1$ classifies a single-feature binary dataset (leftmost plot), i.e. $\alpha \in \mathbb{R}, x \in \mathbb{R}^{N \times 1}$, where initially $\alpha = 0.3$. In the three rightmost plots, the x-axis shows the variation in true (cyan) and surrogate (magenta) losses for the whole space of $\alpha$. The plots indicate that Equation 5 forces the surrogate to approximate the true loss at the regions around the current $\alpha$ (dashed vertical line), while the parameter $\alpha$ is updated towards the minimum of the surrogate by Equation 4.

---

**Algorithm 1:** Surrogate Learning

**Input** : Dataset $\mathcal{D}$, Loss $\ell$, Training epochs $T$, Update steps $K_\alpha, K_\beta$, Learning rates $\eta_\alpha, \eta_\beta$.

1 Initialize $\alpha, \beta$
2 **for** $1, \ldots, T$ **do**
3     $\alpha^{(0)} \leftarrow \alpha$
4     **for** $k = 1, \ldots, K_\alpha$ **do**
5        Sample batch $(x, y) \sim P_\mathcal{D}(x, y)$
6        $\alpha^{(k)} \leftarrow \alpha^{(k-1)} - \eta_{\alpha^{(k-1)}} \nabla_\alpha \hat{\ell}\left(y, \hat{y}\left(x\,;\, \alpha^{(k-1)}\right); \beta\right)$
7     **end**
8     $\alpha \leftarrow \alpha^{(K_\alpha)}, \beta^{(0)} \leftarrow \beta$
9     **for** $k = 1, \ldots, K_\beta$ **do**
10        Sample batch $(x, y) \sim P_\mathcal{D}(x, y)$
11        $\beta^{(k)} \leftarrow \beta^{(k-1)} - \eta_{\beta^{(k-1)}} \nabla_\beta \mathcal{L}\left(\ell\left(y, \hat{y}\left(x; \alpha\right)\right), \hat{\ell}\left(y, \hat{y}\left(x; \alpha\right); \beta^{(k-1)}\right)\right)$
12     **end**
13     $\beta \leftarrow \beta^{(K_\beta)}$
14 **end**

**Return :** Prediction model $\hat{y}(x; \alpha)$

---

in a standard bilevel programming form: $\min_{\alpha \in \mathcal{A}} \hat{\ell}(\alpha, \beta^*(\alpha)), \beta^*(\alpha) = \arg\min_{\beta(\alpha)} \mathcal{L}(\alpha, \beta(\alpha))$. However, the assumptions of the proofs offered by the related work are violated in the case where both considered models $\hat{l}, \mathcal{L}$ are deep neural networks (Franceschi et al., 2018). For example, the assumption necessitating the optimal loss parameters $\arg\min_{\beta(\alpha)} \mathcal{L}(\alpha, \beta(\alpha))$ to represent a singleton for every $\alpha \in \mathcal{A}$ would not hold as there exist continuum many global minima. Nevertheless, we empirically observed that the proposed bilevel optimization problem of Equations 4-5 converges well in practice under the optimization regime of Algorithm 1, visually observable through the exemplifying convergence plots of Section 4.2.

## 4 EXPERIMENTS

### 4.1 PROTOCOL

The experiments are focused on a collection of publicly-available binary classification datasets, whose statistics are presented in Table 1. We split the data randomly into 80% training and 20% testing instances. The prediction model $\hat{y}$ has an architecture of $[100, 30, 10, 1]$ neurons with Leaky-ReLU activation, where batch normalization was applied at each layer. In addition, after each batch normalization we added a drop-out regularization layer with the drop rate set to 0.2. The surrogate network component $g$ has $[30, 30]$ neurons, while $h$ has $[10, 10, 1]$ neurons. We employed the Adam optimizer for training the network, with initial learning rates being $\eta_\alpha = \eta_\beta = 10^{-5}$.

Table 1: Dataset Statistics.

| Dataset | Classes | Train | Test | Features | Pos. Frac. |
|---------|---------|-------|------|----------|------------|
| A9A | 2 | 39073 | 9769 | 123 | 0.2379 |
| CC-Fraud | 2 | 227845 | 56962 | 29 | 0.0015 |
| Cod-RNA | 2 | 390852 | 97713 | 8 | 0.3346 |
| CoverType | 2 | 464809 | 116203 | 54 | 0.4867 |
| IJCNN | 2 | 125344 | 31337 | 22 | 0.0954 |
| Porto-Seguro | 2 | 476169 | 119043 | 57 | 0.0368 |
| Santander | 2 | 160000 | 40000 | 200 | 0.1004 |
| Skin | 2 | 196045 | 49012 | 3 | 0.7919 |
| Susy | 2 | 4000000 | 1000000 | 18 | 0.4577 |

Furthermore, to improve the convergence stability we clipped the gradients by a norm of $10^{-5}$, which is necessary due to the steep curvature of the surrogate losses $\hat{\ell}$, caused by approximating the step-wise true loss surfaces $\ell$. Data batches were drawn in a stratified random fashion (50% positive and 50% negative) with a mini-batch size $N = 100$. The update steps were chosen as $K_\alpha = 3$ and $K_\beta = 10$ and Algorithm 1 was run for 300000 iterations. We conducted experiments with 7 binary classification measures, namely AUC, Equal Error Rate (EER), Average Precision (AP), Miss-classification rate (MCR), F1, Mathew Correlation Coefficient (MCC), Jaccard Coefficient (JAC). All the measures are converted to a loss by a $1 - x$ conversion, except for EER and MCR. For the losses that demand converting the predictions $\hat{y}$ into binary values (e.g. MCR, F1, MCC), we used a thresholding $\hat{y} \geq \gamma$, where $\gamma$ was optimized for each dataset and loss. The following results of Sections 4.2-4.3 present the performance on the test sets.

## 4.2 ABLATION OF SURROGATE MODELS

This section addresses whether the surrogate should be trained in a universal manner (Section 3.1), or whether they should be optimized on a per-dataset basis following the bilevel programming of Section 3.2. For this reason, we designed three different modalities for surrogate learning:

- **Universal Surrogate (SL-U):** Learn the surrogate $\hat{\ell}$ using Equation 3 with hyperparameters $p = 0.5$ (due to stratified sampling), $\mu = 0, \sigma = 1$, and then train only the prediction model $f$ by minimizing $\hat{\ell}$ through Equation 4.

- **Learned from Scratch (SL-S):** Initialize the surrogate network $\hat{\ell}$ from scratch (randomly), then optimize both Equations 4-5 using Algorithm 1.

- **Refined Surrogate (SL-R):** Initialize the surrogate with the universal solution of Equation 3, and then refine the surrogate by optimizing both Equations 4-5 using Algorithm 1.

Table 2 shows the results of the ablation study with the 3 variations of surrogate learning on 7 losses and 9 datasets. We notice that the universal surrogates (SL-U) are overall sub-optimal with respect to the ones trained in a per-dataset manner (SL-R, SL-S), except for the AUC loss. In addition, the results indicate that there is not a major difference in terms of accuracy between SL-R and SL-U. The refined surrogate improves the convergence of the optimization procedure, as Figure 3 shows.

## 4.3 COMPARISON WITH THE STATE-OF-THE-ART

In order to demonstrate the usefulness of learning surrogates, we will compare our method against the state-of-the art baselines of four popular loss functions: MCR, AUC, F1 and JAC. Concretely, cross-entropy is the relaxation of MCR, while Pairwise Ranking (Gao & Zhou, 2015; Chen et al., 2009) and Global Objective (Eban et al., 2017) are the relaxations of AUC. Furthermore, the Lovasz Soft-Max is the relaxation of JAC (Berman et al., 2018; Yu & Blaschko, 2015), and the cost-sensitive reduction (Puthiya Parambath et al., 2014) serves as the surrogate of F1. All the aforementioned baselines, except the cost-sensitive reduction, have no further hyper-parameters and were implemented based on the authors' codes. For a *ceteris paribus* comparison, we used the same capacity prediction

Table 2: Comparing universal surrogates (SL-U) against dataset-specific surrogates (SL-S and SL-R), which are either initialized randomly (SL-S) or with the universal surrogate (SL-R). The lowest loss values are highlighted.

| Dataset | Model | Loss Measures | | | | | | |
|---|---|---|---|---|---|---|---|---|
| | | AUC | EER | AP | MCR | F1 | MCC | JAC |
| A9A | SL-U | 0.0968 | 0.1897 | 0.2629 | 0.2165 | 0.4557 | 0.2164 | 0.1819 |
| | SL-R | 0.0983 | 0.1807 | 0.2584 | 0.1502 | 0.3134 | 0.2085 | 0.1512 |
| | SL-S | 0.0969 | 0.1782 | 0.2585 | 0.1508 | 0.3115 | 0.2069 | 0.1529 |
| CC-Fraud | SL-U | 0.0245 | 0.1512 | 0.2530 | 0.0093 | 0.7774 | 0.3975 | 0.0099 |
| | SL-R | 0.0284 | 0.0914 | 0.1650 | 0.0088 | 0.7693 | 0.3293 | 0.0088 |
| | SL-S | 0.0209 | 0.0814 | 0.1902 | 0.0088 | 0.9970 | 0.3293 | 0.0089 |
| Cod-RNA | SL-U | 0.0107 | 0.0513 | 0.0293 | 0.0510 | 0.1430 | 0.0514 | 0.1272 |
| | SL-R | 0.0110 | 0.0422 | 0.0273 | 0.0430 | 0.0632 | 0.0476 | 0.0426 |
| | SL-S | 0.0101 | 0.0428 | 0.0275 | 0.0418 | 0.0619 | 0.0474 | 0.0422 |
| Covtype | SL-U | 0.1450 | 0.3019 | 0.1730 | 0.3466 | 0.2699 | 0.2349 | 0.2886 |
| | SL-R | 0.1463 | 0.2220 | 0.1663 | 0.2198 | 0.2102 | 0.2165 | 0.2192 |
| | SL-S | 0.1468 | 0.2233 | 0.1635 | 0.2207 | 0.2109 | 0.2203 | 0.2219 |
| IJCNN | SL-U | 0.0048 | 0.0378 | 0.0537 | 0.1045 | 0.5486 | 0.1923 | 0.0609 |
| | SL-R | 0.0030 | 0.0268 | 0.0224 | 0.0161 | 0.0806 | 0.0422 | 0.0152 |
| | SL-S | 0.0028 | 0.0260 | 0.0219 | 0.0153 | 0.0777 | 0.0427 | 0.0161 |
| Porto-Seguro | SL-U | 0.3745 | 0.4348 | 0.9376 | 0.0461 | 0.8990 | 0.4901 | 0.0459 |
| | SL-R | 0.3737 | 0.4110 | 0.9361 | 0.0449 | 0.8867 | 0.4606 | 0.0448 |
| | SL-S | 0.3744 | 0.4127 | 0.9367 | 0.0446 | 0.8939 | 0.4614 | 0.0446 |
| Santander | SL-U | 0.1453 | 0.2349 | 0.5027 | 0.1082 | 0.5800 | 0.3012 | 0.0951 |
| | SL-R | 0.1465 | 0.2300 | 0.4996 | 0.0864 | 0.5166 | 0.2863 | 0.0863 |
| | SL-S | 0.1470 | 0.2314 | 0.4945 | 0.0871 | 0.5083 | 0.2857 | 0.0861 |
| Skin | SL-U | 0.0001 | 0.0648 | 0.0001 | 0.0075 | 0.0036 | 0.0836 | 0.0889 |
| | SL-R | 0.0006 | 0.0044 | 0.0003 | 0.0073 | 0.0085 | 0.0031 | 0.0115 |
| | SL-S | 0.0009 | 0.0143 | 0.0001 | 0.0047 | 0.0017 | 0.0035 | 0.0031 |
| Susy | SL-U | 0.1316 | 0.2269 | 0.1312 | 0.4596 | 0.2507 | 0.2133 | 0.2461 |
| | SL-R | 0.1334 | 0.2156 | 0.1288 | 0.2046 | 0.2289 | 0.2050 | 0.2031 |
| | SL-S | 0.1324 | 0.2140 | 0.1291 | 0.2043 | 0.2294 | 0.2062 | 0.2050 |
| **Ranks** | SL-U | **1.78** | 2.89 | 2.67 | 2.78 | 2.67 | 2.89 | 2.78 |
| | SL-R | 2.22 | 1.67 | **1.67** | 1.78 | 1.78 | **1.44** | **1.56** |
| | SL-S | 1.89 | **1.44** | **1.67** | **1.22** | **1.67** | 1.67 | 1.67 |

Figure 3: Illustrating the convergence for different loss functions on the IJCNN dataset, using two types of surrogates, SL-S: randomly from scratch, SL-R: refined from the universal surrogate. $\hat{\ell}$ and $\mathcal{L}$ represent the surrogate optimization performance on the training set, while $\ell^{\text{Test}}$ the true loss on the tesing set.

model for the baselines, the same batch size, i.e. the same protocol as in Section 4.1. The hyper-parameter of the cost-sensitive reduction, namely the positive weight coefficient was tuned among $\{0.3, 0.9, 2.7, 8.1, 24.3, 72.9\}$ on a separate validation set. To ensure that the baselines converged, we trained them for $1M$ iterations with a learning rate of $10^{-4}$. Table 3 presents the results over the 9 datasets, where the refined surrogate learning SL-R is compared to the 5 state-of-the-art relaxation methods for 4 losses. The evidence suggests that surrogate learning yields more accurate prediction models than the state-of-the-art.

Table 3: Surrogate learning SL-R vs state-of-the-art, **MCR**: CE (Cross-Entropy); **AUC**: PR (Pairwise Ranking (Gao & Zhou, 2015; Chen et al., 2009)), GO (Global Objectives (Eban et al., 2017)); **JAC**: LO (Lovasz Soft-Max for Jaccard (Berman et al., 2018; Yu & Blaschko, 2015)); **F1**: CS (Cost-sensitive F1 reduction (Puthiya Parambath et al., 2014)). Lowest values in bold.

| Data | MCR | | AUC | | | JAC | | F1 | |
|------|-----|-----|-----|-----|-----|-----|-----|-----|-----|
| | CE | SL-R | PR | GO | SL-R | LO | SL-R | CS | SL-R |
| A9A | 0.1520 | 0.1502 | 0.1019 | 0.1028 | 0.0983 | 0.1539 | 0.1512 | 0.3177 | 0.3134 |
| CCF | 0.0088 | 0.0088 | 0.0437 | 0.0369 | 0.0284 | 0.0088 | 0.0088 | 0.7652 | 0.7693 |
| COD | 0.0462 | 0.0430 | 0.0122 | 0.0129 | 0.0110 | 0.0438 | 0.0426 | 0.0652 | 0.0632 |
| COV | 0.2149 | 0.2198 | 0.1786 | 0.1504 | 0.1463 | 0.2594 | 0.2192 | 0.2305 | 0.2102 |
| IJC | 0.0364 | 0.0161 | 0.0168 | 0.0258 | 0.0030 | 0.0322 | 0.0152 | 0.1959 | 0.0806 |
| POR | 0.0446 | 0.0449 | 0.3814 | 0.3815 | 0.3737 | 0.0445 | 0.0448 | 0.8851 | 0.8867 |
| SAN | 0.0842 | 0.0864 | 0.1406 | 0.1427 | 0.1465 | 0.0850 | 0.0863 | 0.4990 | 0.5166 |
| SKI | 0.0482 | 0.0073 | 0.0364 | 0.0473 | 0.0006 | 0.0432 | 0.0115 | 0.0278 | 0.0085 |
| SUS | 0.2146 | 0.2046 | 0.1524 | 0.1508 | 0.1334 | 0.2022 | 0.2031 | 0.2420 | 0.2289 |
| **Wins** | 3.5 | **5.5** | 1.0 | 0.0 | **8.0** | 3.5 | **5.5** | 3.0 | **6.0** |

## 4.4 RUNTIME COMPLEXITY

Denoting the capacities as $\alpha \in \mathbb{R}^{Q_\alpha}, \beta \in \mathbb{R}^{Q_\beta}$, the runtime complexity of Algorithm 1 is $\mathcal{O}\left(T \times \left(K_\alpha \times (Q_\alpha + Q_\beta) + K_\beta \times Q_\beta\right)\right)$, while that of gradient descent for minimizing the cross-entropy (CE) is $\mathcal{O}\left(T \times K_\alpha \times Q_\alpha\right)$. The additive complexity comes from $\frac{\partial \hat{\ell}}{\partial \hat{y}} \frac{\partial \hat{y}}{\partial \alpha}$ where $\mathcal{O}\left(\frac{\partial \hat{\ell}}{\partial \hat{y}}\right)$ is $\mathcal{O}\left(Q_\beta\right)$ in the case of surrogate learning. When deploying on an Intel Xeon E5-2670 server with 40 cores, it took SL-R circa 26 hours and 38 minutes to train the prediction model of Section 4.1 with the MCR loss on the Susy dataset for 1M batches. Under an identical setup, the cross-entropy baseline completed in circa 6 hours and 17 minutes.

## 5 CONCLUSION

The optimization of losses is a major challenge for the machine learning community. Unfortunately, most classification loss functions are only piece-wise continuous, non-differentiable and non-decomposable. So far, researchers have addressed this bottleneck by designing (hand-crafting) smooth approximative surrogate functions to those losses. In contrast to the existing work, we propose a new paradigm to optimizing loss functions, by defining the loss itself as a parametric model that is jointly optimized with a prediction model, in a way that the smooth surrogate loss matches the non-differentiable true loss. The task is formalized as a bilevel programming objective and an alternating optimization algorithm is applied to learn the surrogates. The empirical results on multiple real-life datasets indicate that learning surrogates is more accurate than hand-crafted explicit relaxations in diverse popular loss functions, such as AUC, F1, or Jaccard Index.

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
