# OpenReview forum: "Learning Surrogate Losses"
_ICLR.cc/2020/Conference — Reject_

### Official Review · AnonReviewer2 · 2019-10-22
**Official Blind Review #2**

**Rating:** 3

**Review:**

This paper proposes a method of learning loss functions in addition to the learning of predictors. Since it's not easy to optimize loss functions that evaluate the accuracy, surrogate loss functions have been widely employed. The design of the surrogate loss is problem-dependent, and handcraft is required. This paper tries to tackle this problem from the viewpoint of meta-learning, i.e., the surrogate loss learning. Typically, deep neural networks (DNN) are used to design a surrogate loss that approximates the original loss while maintaining the tractability of the optimization. Some convergence properties of the proposed method are analyzed. Some empirical studies showed the efficiency of the proposed method to the state-of-the-art baselines.

The design of the surrogate loss is important for machine learning problems. However, the proposed method in this paper seems an ad-hoc approach rather. For example, the 0-1 loss is often replaced with convex loss functions such as the hinge loss or logistic loss. Using these surrogate loss functions, the statistical properties of the predictors obtained from 0-1 loss are maintained. See the following paper for details.
P. L. Bartlett, et al., (2006), Convexity, Classification, and Risk Bounds, Journal of the American Statistical Association March , Vol. 101, No. 473.

On the other hand, the current approach does not have such a theoretical guarantee for each learning problems. Though certainly, the proposed method is widely applicable to many problems, there is no theoretical guarantee. Theorem 1 in page 5 shows the convergence property. However, the number of iterations, K_beta, should tend to infinity. This is not a practical operation in the learning algorithm

**Experience Assessment:**

I have published in this field for several years.

**Review Assessment: Checking Correctness Of Derivations And Theory:**

I did not assess the derivations or theory.

**Review Assessment: Checking Correctness Of Experiments:**

I did not assess the experiments.

**Review Assessment: Thoroughness In Paper Reading:**

I made a quick assessment of this paper.

---

> ### Author Response · Authors · 2019-11-06
> **Clarifications to Reviewer 2**
>
> Thanks for the important point you raised. We would like to address both aspects of your review:
>
> a) The proposed surrogate loss model is ad-hoc
> b) There is no theoretic guarantee of the surrogate loss in approximating the true loss
>
> Regarding point a)
>
> Please let me remind that given the true targets $y \in \mathbb{R}^N$ of a batch having $N$ instances and the estimated targets $\hat y \in \mathbb{R}^N$ of that batch, then a loss function is defined as the true loss $\ell(y, \hat y): \mathbb{R}^{2N} \rightarrow \mathbb{R}$. The purpose of a surrogate $\hat \ell(y, \hat y)$ is, therefore, to approximate well such a true loss $\ell(y, \hat y)$.
>
> The first naive intuition would be to define $\hat \ell(y, \hat y)$ as a plain MLP neural network, which by virtue of the universal approximation theorem is guaranteed to estimate $\ell(y, \hat y)$. However, further thinking reveals that $\hat \ell$ cannot be a plain neural network with $2N$ inputs and $1$ output, because of the permutation-invariant treat of the true losses. Notice that the loss output should remain the same even if the order of the instances within a batch is shuffled (a.k.a. permutation-invariant). A neural network, on the other hand, does not ensure that the output remains the same if the order of features is swapped.
>
> Given that loss functions are permutation-invariant, we came up with the intuition that a loss is actually a mathematical function that operates over sets. In this case the sets of $(y, \hat  y) \in \mathbb{R}^2$ values for a set of batch instances. The set formulation of the loss handles the permutation-invariant nature of randomly shuffling instances within a batch. A recent work [1] has actually proven that functions expressed via the Kolmogorov-Arnold representation theorem are permutation-invariant. As a result, the surrogate loss we propose in Equation 2 is actually the methodologically-correct surrogate model to approximate the permutation-invariant true loss functions.
>
> Regarding point b)
>
> We would like to clarify that the Kolmogorov-Arnold representation is well-known to offer theoretical guarantees in terms of approximating the target function [3]. Even in the concrete case where the functions $g$ and $h$ of Equation 2 are ReLU networks (as in our paper), the respective error bounds exist and are derived by prior literature [2].
>
> As a result, since our surrogate network is an instance of the standard Kolmogorov-Arnold representation, it automatically benefits from the known aforementioned theoretical guarantees of the Kolmogorov-Arnold representation theorem.
>
> We mentioned this aspect briefly in Section 3, however, thanks to your comment we will expand the clarification with a paragraph on its own for the rebuttal.
>
> -----------
>
> Regarding the comment on $K_\beta$ tending to infinity in the optimization algorithm, please notice that in practice the models converge with a small $K_\beta=10$ (Section 4.1). We can actually observe the convergence in Figure 3 of Section 4.2, and the fact that the converged models outperform the state-of-the-art in Section 4.3. We would like to point out that these types of bi-level optimization algorithms are known to converge and also applied to other domains, such as Hyper-parameter optimization [4].
>
> -----------
>
> To recap:
> * the proposed surrogate loss model is the methodologically-correct approach to handle the permutation-invariant nature of the true loss,
> * theoretical guarantees are inherited by the properties of the well-studied Kolmogorov-Arnold representation theorem.
>
> Please let us know if our clarification does not fully address your concerns?
>
> [1] Zaheer et al., Deep Sets, NIPS 2017
> [2] Montanelli et al., Error bounds for deep ReLU networks using the Kolmogorov–Arnold superposition theorem, ArXiv 2019, https://arxiv.org/pdf/1906.11945.pdf
> [3] Zhang, Kolmogorov’s Superposition Theorem, 2006, https://www.maths.ed.ac.uk/~xzhang/files/oct_2016_xiling.pdf
> [4] Franceschi et al., Bilevel Programming for Hyperparameter Optimization and Meta-Learning, ICML 2018

---

### Official Review · AnonReviewer1 · 2019-10-23
**Official Blind Review #1**

**Rating:** 8

**Review:**

In this paper authors propose to jointly optimize a critic, that estimates some non-differentiable objective (or an objective with intractable derivatives and/or derivatives that are 0 almost everywhere). Authors conduct numerous experiments, and show improvements over some sota methods, and maybe more importantly - provide a unified way of achieving these across multiple metrics. Paper is well written, very easy to understand and in reviewers opinion, a nice, simple story worth sharing.


Major concerns:

The theoretical result provided is just a citation of an existing proof, rather than a new contribution. More importantly however, the setup considered strongly violates the assumptions of the theorem, for example assumption (b) does not hold for the neural networks, as there are continuum many global minima (imagine having a unit in a neural net, that has a huge negative bias, which causes it to always be "turned off" by relu, then any weight on top of it will have the same final loss value; or simply notice that relu(a*x)=a*relu(x) if x>0, and so you can always "push" the norm of weights from one layer to the next one without affecting the outcome etc.), and so argmin is not a singleton, but the set of exactly same power as the whole parameter space (since for each k there exists a bijection from R^k to R, and R is of continuum power). This property is not a mild assumption, but rather a critical element guaranteeing convergence of such systems, with powerful universal approximators, one cannot hope for such strong convergence results without actually analysing the approximators family. Reviewer strongly believes this should be either removed completely, or just briefly mentioned, rather than made a strong statement in the paper. It is an empirical work, and stands strong on its own rights, there is no need to add theorems with assumptions that are never satisfied in the proposed scenario.

Results provided use somewhat non-standard datasets for deep learning, and as such it is hard to asses statistical significance of the differences reported; while the test datasets sizes are big enough to trust error to 1e-3 level, they are heavily imbalanced and relatively low dimensional problems, which have proven to be hard for neural network many times in the past, consequently I would expect to see confidence intervals or stds of each result, at least for Table 3. The problem becomes even more severe for metrics such as F1 which do not decompose additively and so can be very sensitive to the (false) positive rates values - reported improvements might disappear once these are introduced, but even if one does not outperform hand-crafted proxies for specific objectives, having a unified method that is on-part with those in black-box scenario is a good result (I would argue that even if all the results are slightly worse, it is still worth publishing).

I find it a bit disappointing, that authors did not try to analyse trained models, it would be invaluable to see what kind of aggregation g and h came up with, that is well aligned with losses such as F1 or JAC.


Minor concerns:

The work resembles closely methods of error critic learning - where one uses an update direction coming from a model, that regresses towards the loss itself, or the gradient of the loss (is available), see:
- Neural Network Design for J Function Approximation in Dynamic Programming (Pang and Werbos paper from '98)
- Sobolev Training of Neural Networks (from NIPS; more precisely "Critic" baseline, which has the same functional form as the loss presented in the paper, if applied at the very top of the network only)
It might be worth discussing in the paper.


Overall I recommend weak acceptance, and encourage authors to address some of the concerns raised above.

**Update**

Based on discussion, corrections and new additions, I recommend acceptance of the above work.

**Experience Assessment:**

I have published in this field for several years.

**Review Assessment: Checking Correctness Of Derivations And Theory:**

I carefully checked the derivations and theory.

**Review Assessment: Checking Correctness Of Experiments:**

I assessed the sensibility of the experiments.

**Review Assessment: Thoroughness In Paper Reading:**

I read the paper thoroughly.

---

> ### Author Response · Authors · 2019-11-07
> **Clarifications to Reviewer 1**
>
> Thanks a lot for reading the paper in-depth and providing valuable criticism.
>
> Regarding the major concerns:
>
> * We agree with your reasoning that the singleton assumption is too strong, therefore we accept the criticism. Your suggestion to shorten the convergence note will be taken into consideration. As a result, we will reduce that section to a few sentences in the rebuttal to make room for addressing the comments of the reviewers.
>
> * Related to the choice of datasets, we tried to blend a mix of UCI datasets with more modern datasets from Kaggle. Please notice that the imbalance of the datasets is actually a desirable property rather than a bottleneck because in such datasets it more suitable to assess the performance of metrics such as AUC, F1, etc.. On the other hand, the imbalance is obviously not a facultative option, but rather a real-life challenge. Since the experimental protocol was conducted on a random 80%-20% split, the confidence intervals are not defined. Yet, we agree that the size of the test sets is large enough to offer trustable empirical evidence.
>
> * On the other hand, you are right in stating that optimizing directly for F1 or AUC in such imbalanced datasets gives a lift if we would have compared against balance-sensitive decomposable losses such as cross-entropy. However, please notice that we actually compared and outperformed the state-of-the-art baselines of Section 4.3 which also directly optimize for these imbalance-handling losses (such as AUC, F1, Jaccard).
>
> * Yet again, we agree that the interpretability of the surrogate losses would be nice to have. Unfortunately, it is not trivial to analyze how the Q=30 latent per-instance error components of function g directly relate to, for example, the four confusion matrix measures of F1. Even if we force Q=4 there is no guarantee that the latent error components would directly match the confusion matrix, since the multi-layered aggregation network h combines the components of g in a black-box (a.k.a. neural network-ish) manner, and not in the same way that common losses (such as F1) do. Perhaps one can extend this work by a multi-target surrogate optimization where we set Q=4 and at jointly i) train the output of g to estimate the confusion matrix terms, and ii) train the output of h to match the final metric (e.g. F1, Jaccard).  The disadvantage of this approach is that it would not be directly applicable to AUC, because we have multiple confusion matrices for the various thresholds used to binarize the estimated target. As a result, it actually seems not straightforward to analyze the black-box surrogates and interpret their neural networks.
>
> Regarding Minor concerns:
>
> The relation you pointed out to error critic learning seems relevant and we will incorporate it into the related work.
>
> However, I failed to see the relation to Sobolev training, In our case, there is no information on the gradients of the target loss with respect to the activations of the different layers of the surrogate model, since in contrast to the assumptions of the NIPS paper our target loss function has no differentiable form.

---

> > ### Comment · AnonReviewer1 · 2019-11-08
> > **Thanks for the response**
> >
> > Thank you for the detailed response.
> >
> > In terms of datasets - I just want to make clear that I was not criticising the choice of datasets, but rather seeking some statistical significance confirmation. The imbalance is indeed the crucial real-life phenomenon, but it affects sensitivity of some metrics to noise - for example every balanced metric can be heavily changes by tiniest change in correctness of minority class accuracy, and as such - more than one run, and corresponding std dev/std err reporting. While authors claim to outperform the SOTA methods there - this is exactly something I would prefer to see verified with at least some notion of statistical significance.
> >
> > In terms of analysis it is of course tricky to know what neural networks do represent, however there are various hypotheses one could put forward and try to verify. For example - can we decode typical metrics, such as number of FP, TP, FN, TN from the internal representation? These questions can be answered by trying to fit a simple decoder (even linear one) on top of representation learned, and doing a train-test split among the datasets/samples to see if this is indeed the case.
> >
> > Finally - Sobolev training paper was only mentioned wrt. use of critic learning, not as a technique that has something specific to do with the paper under consideration, sorry for the confusion.

---

> > > ### Author Response · Authors · 2019-11-08
> > > **On statistical significance**
> > >
> > > Thanks for the time responding to our earlier comments.
> > >
> > > A k-fold cross-validation will demand a k-times repetition of all the experiments, which poses a challenge given the remaining time until the rebuttal deadline.
> > >
> > > Luckily, there exists a popular test to compute the significance of the single-split results of two algorithms across different datasets: the Wilcoxon Signed-Rank Test [1].
> > >
> > > We ran the two-tailed hypothesis variant with a significance level of p=0.05 (standard setup). Practically speaking: the test operates using pairs of values from every two columns (SL-R and baseline) of Table 3. We used the test based on the z-value analysis as the prior literature [1] recommends. The test was conducted through https://www.socscistatistics.com/tests/signedranks/default2.aspx . We present below two modes of conducting the test i) Aggregating across datasets and ii) Aggregating across both datasets and losses
> > >
> > > i) Aggregating across datasets:
> > >
> > > The first test is the aggregation of values across datasets, which is feasible since the results on each dataset measure the same range of loss values between [0,1] with 0 being the perfect test score. The website was programmed to allow the computation of the z-value for a minimum of 10 pairs of values (not sure why this limit), while we have 9 datasets. To overcome this technical limitation we entered each row/dataset twice to yield 18 values and overcome the 10 values limit. The p-value results based on the z-value analysis were as follows:
> > >
> > > SL-R vs CE: p=0.0703
> > > SL-R vs PR: p=0.0012
> > > SL-R vs GO: p=0.00062
> > > SL-R vs LO: p=0.02642
> > > SL-R vs CS: p=0.03486
> > >
> > > Interpretation of the findings:
> > > The significance w.r.t the AUC baselines is quite strong as the p-value goes towards zero. The significances w.r.t. the F1 baseline (CS) and the Jaccard baseline (LO) pass the Wilcoxon 5%-significance test, albeit less significantly than AUC (even though p values are less than 0.05 they are considerably higher than 0). Unfortunately, our method is not significantly better than the Miss-classification Rate baseline (Cross-Entropy) (p=0.07 does not pass the 0.05 test).
> > >
> > > ii) Aggregating across both datasets and losses:
> > >
> > > Such a test will compare all the scores of SL-R across all datasets and losses, by pairing values with the respective baseline of each specific loss (i.e. there are 36 pairs of values for SL-R vs each of the 4 baselines in 9 datasets). Such aggregation is again feasible since each value measures the same range of the loss values between [0,1], with 0 being the perfect test loss.  Notice that since AUC has two baselines, PR and GO, we repeated the test twice with each of the alternatives. The other losses have only a single baseline. The statistical significance results are as follows:
> > >
> > > SL-R vs Baselines (PR for AUC): p=0.00072
> > > SL-R vs Baselines (GO for AUC): p=0.00062
> > >
> > > Interpretation of the findings:
> > > When comparing our method to all the baselines, it results that the Wilcoxon signed-rank test is quite conclusive that the difference is statistically significant (p values are way smaller than 0.05 and close to 0). The conclusive message of the test is that the improvement over baselines across datasets and losses cannot be attributed to split randomness under a 5% significance.
> > >
> > > Do you find this evidence sufficient regarding the significance?
> > >
> > > [1] Demsar, Statistical Comparisons of Classifiers over Multiple Data Sets, JMLR 2006

---

> > > > ### Comment · AnonReviewer1 · 2019-11-11
> > > > **On statistical significance and analysis**
> > > >
> > > > I would like to thank authors for taking time to provide additional analysis and validation. It is understandable that k-fold validation might be too expensive, and while I would recommend, for future work, to rather provide multiple trials on a smaller set of datasets, than limited number of trials on many datasets (since the first one is more closely related to actual test domain, which is other researcher using proposed method on a single dataset/problem of interest), I view current results as sufficient regarding the significance levels.
> > > >
> > > > To add a few notes to previous comments on analysis, apart from trying to decode metrics from internal representation, there are other small-ish experiments that could be run to understand what the networks have learned (for example - doing these experiments on just one dataset, where results are really good, instead of doing full sweep):
> > > > - removing "h" from the equation (substituting it with "reduce mean" operation), and only learning g, to show whether truly the learned model is non decomposable to sum objective (the fact, that "true" loss is not decomposible, does not mean that surrogate one ended up with, isn't) This would allow to strengthen the claims about non-decomposability (if the answer is positive) or provide cool counter-intuitive result of existence of decomposable surrogates (if answer is negative) - note that this can still be true in adaptive setup - since surrogate loss is trained with the network, it might be decomposible in each iteration, while not being "globally" decomposible
> > > > - for the losses that actually are "well behaved" - such as CE, it would be interesting to see whether the surrogate shares any important properties of the underlying loss - does it match the derivatives? Is the surrogate loss convex? This could help understand whether learned surrogates are replicating the true loss, or rather look for new ways to minimise the function without following the gradient (and thus could suggest that the field, which arguably is a bit "gradient obsessed" might want to look into different dynamical systems)
> > > > - once the surrogate loss is adapted, can it be used to train on this dataset again, from scratch? This would help to disentangle, whether the "fine tuning" stage is "just" fine tuning the global geometry, or it actually has to be an adaptive process, and effectively what one gets is a very local model, that only works around current \theta, rather than whole parameter space
> > > >
> > > > Note, that neither of these suggestions is criticism of the work done, but rather encouragement to provide readers with more insights, into very interesting project and results that your group obtained.

---

> > > > > ### Author Response · Authors · 2019-11-13
> > > > > **On latest suggestions**
> > > > >
> > > > > Thanks a lot for the nice suggestions, our comments are as follows.
> > > > >
> > > > > Part 1 - Non-decomposability Experiment
> > > > >
> > > > > It is possible to systematically structure your suggested experiments as an ablation of the capacity of $h(x)$ where $x = \frac{1}{N} \sum\limits_{i=1}^{N} g(y_i, \hat y_i)$
> > > > >
> > > > > Case 1: SL-Deep, where $h(x) = MLP(x; \alpha)$ with 3 layers and layer sizes [10, 10, 1] as in the paper
> > > > > Case 2: SL-Lin, where $h(x) = \langle x, \alpha \rangle + \alpha_0$ is a linear function, i.e. a one layer MLP with a single neuron, layer size [1] (a.k.a. the decoder suggestion)
> > > > > Case 3: SL-Ident, where $h(x) = x$ is an identity function (a.k.a. the reduce_mean suggestion)
> > > > >
> > > > > The results are run on the IJCNN1 dataset testing the three cases with the MCR, AUC and F1 losses. For all cases the same protocol was used, same learning rate, batch size, number of epochs as detailed in Section 4.1.
> > > > >
> > > > > Loss, SL-Deep, SL-Lin, SL-Ident
> > > > > AUC, 0.0029, 0.0035, 0.0033
> > > > > MCR, 0.0153, 0.0157, 0.0185
> > > > > F1, 0.0796, 0.0793, 0.0868
> > > > >
> > > > > Overall it seems that a non-identity deep aggregation $h$ helps, but is not very clear why does an identity aggregation performs slightly better than a linear model on the AUC loss. Also, it is a counter-intuituve why does a deep $h$ outperform the identity $h$ in the case of the Miss-classification Rate (MCR) knowing that MCR is decomposable. Hypothetically, the capacity of the per-instance error representations network $g(y, \hat y)$ might need to be higher in the case of SL-Ident, since the other parametric versions of $h$ are fitting the surrogate loss better on the training set (e.g. train loss for MCR were SL-Deep: 0.0137, SL-Lin: 0.0128, SL-Ident: 0.0158), i.e. removing $h$ deteriorates the fit of the identity aggregation.  Nevertheless, all ablations of $h$ outperform the SOTA models of Section 4.3 in IJCNN1 for the tested losses MCR, AUC and F1.
> > > > >
> > > > > Part 2 - Interpretation of the learned surrogate
> > > > >
> > > > > On the question whether the learned surrogates share properties of the underlying loss:
> > > > > * Due to being a standard neural network, the surrogate loss is not convex, neither with respect to the estimated targets, nor with respect to the prediction model weights. On the other hand the cross-entropy is convex with respect to the estimated target, but equally non-convex with respect to the prediction model weights.
> > > > > * Our method is also "gradient-obsessed" to follow your metaphor, since steps 4-7 of Algorithm 1 also update the prediction model parameters $\alpha$ with respect to the gradient $\frac{\partial \hat \ell}{\partial \alpha}$ of the surrogate $\hat \ell$.
> > > > > * However, there is a major difference that I would like to emphasize with the help of Figure 1. Notice that the surrogate loss $\hat \ell$ app (magenta line) approximates the true loss $\ell$ (cyan line) at the parameters $\alpha$ it has observed (vertical dashed black line). This is because lines 9-12 of the algorithm update the parameters $\beta$ of $\hat \ell$ for the actual $\alpha$. As a result the surrogate loss is a local approximation of the true loss only in the explored space of the prediction model parameters $\alpha$.
> > > > >
> > > > > I did not fully understand the last suggestion with the adaptation and the global geometry. Please feel free to expand that point if it is not already covered by the previous comments.

---

> > > > > > ### Comment · AnonReviewer1 · 2019-11-14
> > > > > > **Re: latest suggestions**
> > > > > >
> > > > > > I would like to thank authors again for quick responses, and providing additional results and analysis.
> > > > > >
> > > > > > Part 1
> > > > > >
> > > > > > The ablation showing that the trained model indeed requires a custom aggregation is interesting, and if authors have time to show this with repeated experiments for the camera ready version of the paper - it would be a nice addon.
> > > > > >
> > > > > > Part 2
> > > > > >
> > > > > > In terms of geometric properties I mostly meant whether these qualities (e.g. convexity) are obeyed at all, of course there is no constraint like this in your current model, so we can't expect to have full convexity, but one can use monte carlo methods to estimate "how non-convex" is it.
> > > > > >
> > > > > > Part 3
> > > > > >
> > > > > > This suggestion was simply to ask, if once fully converged, fine tuned version of your model, can be used to train on the same dataset but with random initialization. One can think about at least a few hypotheses why fine tuning (co-training of loss and model) works well:
> > > > > > 1. There is something special about global geometry of this loss on this dataset, fine tuning allows to express it, and once it is done - it can be used to train from scratch
> > > > > > 2. The effect has purely local property - loss surface is so complex, that proper approximation can only happen around current theta, at the cost of "forgetting" how the function looks like at theta_0 etc. So if we were to freeze the loss net after fine-tuning and reinitialize the model - it would not perform well.
> > > > > > Either result would be interesting, and it is not something that would (in)validate the method, but rather would provide readers with more intuitions/insights into why is it working well. Given limited time I would encourage authors to consider doing this for camera ready version.
> > > > > >
> > > > > > Overall I am satisfied with the modifications and corrections authors did, and would recommend acceptance of the paper.

---

### Official Review · AnonReviewer3 · 2019-10-24
**Official Blind Review #3**

**Rating:** 3

**Review:**

In this paper, the authors propose to learn surrogate loss functions for non-differentiable and non-decomposable loss. An alternative minimization method is used for training the surrogate network and prediction model. Learning surrogate loss functions for different tasks is somewhat novel, although there are some prior works on learning the loss, e.g., [1].

Pros.
1.	The paper is well written and easy to follow.
2.	Learning a unified loss for different tasks is interesting. It has the potential to reduce human efforts to design losses.

Cons.
1.	Some important details of learning the surrogate loss are missing. The function g for extracting the latent error component is not clear. The composition function h is not provided either.
2.	The details of the experiments are not clear. Are the results on the test set? How many independent runs are performed?
3.	Learning surrogate loss incurs additional approximation error, time complexity, and model complexity. The benefit of the trade-off is not systematically evaluated.

Questions
1.	The function g for extracting the latent error component is not clear. Is it required to design for different tasks by an expert specifically? Or, does it have the same differentiable form for different tasks? Please provide the details of the function g for the different losses in the experiments.

2.	Can the proposed loss work well for multi-class classification tasks? In the experiments, only binary classification is evaluated. Multi-class classification will increase the number of classes, thus increasing the difficulty of approximation. It is better to provide MCR compared with CE on the CIFAR100 dataset for evaluation. Also, please provide the running time of CE and SL-R in the same running environment.

3.	Is the results in Table 3 on the test set? CE has fewer parameters compared with the proposed loss, why does SL-R have better generalization performance compared with CE? How many independent runs performed in experiments?

4.	 CE does not need training the loss compared with SL-R. Please provide the running time of CE in the same environment for a fair comparison.

5.	The time complexity analysis treats extracting function g as a black-box function. However, the complexity of function g depends on the tasks. Please provide a detailed discussion about time complexity for different tasks (e.g., AUC, F1, MCR for multi-class classification, and ranking tasks).


[1] Learning Loss Functions for Semi-supervised Learning via Discriminative Adversarial Network, 2017



**Experience Assessment:**

I have published one or two papers in this area.

**Review Assessment: Checking Correctness Of Derivations And Theory:**

I did not assess the derivations or theory.

**Review Assessment: Checking Correctness Of Experiments:**

I assessed the sensibility of the experiments.

**Review Assessment: Thoroughness In Paper Reading:**

I read the paper at least twice and used my best judgement in assessing the paper.

---

> ### Author Response · Authors · 2019-11-05
> **Clarifications to Reviewer 3**
>
> Thanks a lot for your constructive comments. I would like to address each concern in the same order.
>
> First of all thanks for reference [1], we will cite and address it.
>
> Regarding Cons.:
>
> 1. The function g and h are specified in Section 3, the fourth paragraph of page 3 to be deep-forward networks (MLP). The architectures for g and h are detailed in Section 4.1. Notice that the architecture of $g$ & $h$ is the same for all losses, but the weights of $g$ & $h$ are trained specifically for each loss $\ell$. The function $g$ and $h$ are represented by the parameters $\beta$ in Equations 4-5 and Algorithm 1. Our method automatically fits the surrogate to each task via an end-to-end optimization.
>
> 2. The reported results are measured on the test set following the protocol of Section 4.1.
>
> 3. The benefit of our method is an end-to-end surrogate loss, whose optimization improves the state-of-the-art in terms of multiple losses (Section 4.3). The only disadvantage is an added run-time component in computing the gradient of the loss with respect to the estimated target. Please refer to Section 4.4. for additional run-time analysis.
>
> Regarding Questions:
>
> 1. The details for the functions g and h are already provided in Section 3 and Section 4.1 as MLP networks. No expert hand-crafting of those networks are needed, on the contrary, the process is an end-to-end learning approach conducted via Algorithm 1. We showed in Section 4 that a common architecture design for g and h is sufficient to advance the state-of-the-art in terms of popular losses, such as AUC, F1, Jaccard, etc.
>
> 2. Please notice that we already experimented with 7 different non-differentiable measures and 9 datasets. In our review, no other published paper has actually experimented with a larger number of non-differentiable losses. On the other hand, we wanted to exclusively focus on binary classification first in order to systematically and exhaustively analyze this setup, then afterwards extend the work with future papers in different areas, e.g. losses for recommender systems (MAP, Hit-rate@K), computer vision (top-k multi-class accuracy, IoU bounding box), or language translation (BLEU). However, we believe binary classification losses are valuable to the Machine Learning community with implications in imbalanced classification (AUC, F1), biometric verification (Equal Error Rate), etc, which justifies the experimental setup.
>
> 3. The results are presented on the test set following the protocol of Section 4.1. The advantage of our approach compared to surrogates like CE is that a parametric surrogate fits the true loss better in the loss regions dictated by the current prediction model. In other words, the surrogate loss params beta are fitted for the actual alpha in a dataset-specific manner (see Section 3.2, Equation 5), while non-parametric surrogates are not tailored to dataset-specific relevant regions of the loss surface.
>
> 4. The question on the runtime figures is answered in a separate message.
>
> 5. Please notice that function g is not treated as a black-box in the runtime analysis. The weights of g and h are represented by the parameters vector $\beta$ whose dimensionality is defined as $Q_\beta$ and is used in the asymptotic order.  On the other hand, we agree that the complexity of a surrogate is dependent on the target loss. In fact, we used the same architecture of g and h for all target losses (AUC, F1, MCR, etc.) and did not search the architectures on a per-loss basis, in order to keep the experiments feasible under our computational resources. However, it turned out that a common surrogate network architecture was actually sufficient to outperform the state-of-the-art surrogates of all target losses (Section 4.3). After further thinking we concluded that it causes no harm to have a common deep surrogate loss network with a sufficiently high capacity for all target losses, since in the worst case, a very large capacity surrogate will fit the non-differential losses better and might have steeper loss surfaces than a small capacity surrogate around the step-like surfaces of the true loss (see Figure 1 for the step-like loss surfaces). However, if one uses gradient clipping as we do, the gradients of the surrogate loss wrt the estimated target variable will not explode at the steep loss surface regions of the high capacity surrogate loss surface. In the end, the very good empirical results compared to the state-of-the-art supported our hypothesis. Therefore, given a common g and h architecture, the time complexity in our setup remains the same independent of the target loss.
>
> As a result of your helpful comments, we will update the paper accordingly to emphasize the answers to the raised questions.
>
> Please, let me know if something is not clear, or any further input is needed.

---

> ### Author Response · Authors · 2019-11-11
> **Regarding Point 4: Runtime**
>
> We finished computing your demanded runtime figures on comparing CE vs SL in the same environment. The largest dataset Susy was selected, having with 4M training instances, 1M testing instances, and 18 features. For both losses, we trained the same model (Section 4.1) with exactly the same configuration (batch size 100, number of batches 1M). The experiments were run on the same Intel® Xeon® Processor E5-2670 v2 servers with 40 cores.
>
> CE took ca. 6 hours and 17 minutes to train for 1M batches
> SL took ca. 26 hours and 38 minutes to train for 1M batches
>
> The figures indicate that SL is less than an order of magnitude slower (~4x slower) than the fast and non-parametric cross-entropy loss. Such findings are also compliant with our asymptotic runtime analysis of Section 4.4.
>
> Therefore, the aforementioned runtime cost is the trade-off that you correctly mentioned in your review.
>
> Please let us know if any additional concerns remain.

---

### Author Response · Authors · 2019-11-15
**Updates for the Rebuttal**

Thanks a lot to all the reviewers for providing new insights and valuable positive criticism of our work.

As a consequence, we updated our paper to emphasize, among other minor points, the following key review aspects:

* Reviewer 3 correctly mentioned the trade-off (disadvantage) aspects of the surrogate models, with respect to the run-time. We conducted a fair run-time experiment to the baseline and updated Section 4.4.

* Reviewer 1 rightfully criticized the assumptions of the converge proof of Section 3.3, which was amended and shortened according to her/his suggestion. The statistical significance results were also presented in our discussions, in addition to a new ablation experiment on the capacity of per-instance score aggregation network.

* Reviewer 2 legitimately questioned the theoretical guarantees of the error bounds of the surrogate model in approximating the true losses. As a result, a clarification of the existing guarantees which are inherited by the Kolmogorov-Arnold representation theorem was added in Section 3.

- In addition, the related work was expanded with the suggested literature of Reviewers 3 and 2.

Besides the aforementioned points, we commented on the raised questions in this forum.

---

### Decision · Program_Chairs · 2019-12-19

**Decision:**

Reject

**Comment:**

Unfortunately, this was a borderline paper that generated disagreement among the reviewers.  After high level round of additional deliberation it was decided that this paper does not yet meet the standard for acceptance.  The paper proposes a potentially interesting approach to learning surrogates for non-differentiable and non-decomposable loss functions.  However, the work is a bit shallow technically, as any supporting theoretical justification is supplied by pointing to other work.  The paper would be stronger with a more serious and comprehensive analysis.  The reviewers criticized the lack of clarity in the technical exposition, which the authors attempted to mitigate in the rebuttal/revision process.  The paper would benefit from additional clarity and systematic presentation of complete details to allow reproduction.